# A Challenging Nodular Lesion of the Ear

**DOI:** 10.3390/medicina58020269

**Published:** 2022-02-11

**Authors:** Antonella Tammaro, Carmen Cantisani, Camilla Chello, Ganiyat Adenike Ralitsa Adebanjo, Lavinia Lilli, Francesca Farnetani, Chiara Filippi, Edoardo Covelli, Evelina Rogges, Giovanni Pellacani

**Affiliations:** 1NESMOS Dermatology Department, Sapienza Medical School of Rome, 00141 Rome, Italy; antonella.tammaro@uniroma1.it (A.T.); camilla.chello@gmail.com (C.C.); g.adebanjo@gmail.com (G.A.R.A.); e.covelli@gmail.com (E.C.); 2UOC of Dermatology, Policlinico Umberto I Hospital, Sapienza Medical School of Rome, 00161 Rome, Italy; farnetani.francesca@gmail.com (F.F.); giovanni.pellacani@uniroma1.it (G.P.); 3Dermatology Department, University of Modena and Reggio Emilia, 41124 Modena, Italy; 4Biological Science Department, “Roma Tre” University, 00154 Rome, Italy; l.lilli@gmail.com; 5ENT Department, Sapienza University of Rome, 00161 Rome, Italy; c.filippi@gmail.com; 6Pathology Department, Sapienza University of Rome, 00161 Rome, Italy; e.rogges@gmail.com

**Keywords:** ear, Merkel cell carcinoma, nodular lesion

## Abstract

Skin nodular lesion are really frequent, but rapidly growing ones needs to be quickly removed since they can hide really aggressive skin tumor. Among malignant lesion Merkel cell carcinoma arise. It is a rare neuroendocrine skin tumor highly aggressive, not easy to diagnose at first stage, since at first diagnosis it is already widespreading all over the body. In order to renew interest in this letal skin tumori is mandatory to remind high risk population which include elderly people, white skin, chronically exposed to UV immunocompromised. Our unhappy case was described to increase awareness on this kind of skin tumor, since new drug appeared in the market can give an hope to these patients.

## 1. Introduction

Cutaneous nodular lesions have always been challenging due to their clinical and sometimes dermoscopic similarities, which leads to an even more complex diagnostic process when they display more aggressive behavior. Although frequently related to malignant tumoral lesions, benign lesions may also present as growing nodules [1,2,3]. The location of the lesion may create additional challenges: if localized on areas such as the head and neck [3], the diagnostic incisional biopsy may leave aesthetically unpleasant scars which could not be accepted by the patient, especially if the lesion were to be of a benign nature. Among malignant nodular lesions most commonly observed are basal cell and squamous cell carcinoma and all their histopathological variants, as well as melanoma, metastasis, neurofibromas, lymphoma, and Merkel cell carcinoma. The clinical appearance, evolution, color, and age of patient can be useful information in making a diagnosis.

We describe a case of an old patient with a rapidly growing violaceous lesion.

## 2. Case Report

A 75-year-old man with a personal history of bipolar disorder, diabetes, and smoking was admitted to the Dermatologic department because of a recent cutaneous nodular lesion appearing on the top of his right ear.

On physical examination, a violaceous nodule on the right helix was observed (Figure 1). The dermoscopic aspect was characterized by polymorphic vascular pattern The lesion was firm at palpation and the patient did not report any related symptoms.

An incisional biopsy was performed: the histology section showed small roundish blue cells with basophilic nuclei and scarce cytoplasm organized in nests. A high mitotic rate was present and immunohistochemistry showed positivity for CD56 and synaptophysin (Figure 2); thus, the diagnosis of Merkel Cell carcinoma was made.

To investigate local and distant body sites, a total body CT scan was performed, which highlighted the presence of neoplastic tissue in the right lung, with the complete atelectasis of the medial and inferior lobe and compression of the principal bronchus. An investigative biopsy was planned to evaluate the nature of the infiltrative tissue in the lung, but the patient died before it could be performed.

## 3. Discussion

Merkel cell carcinoma (MCC) is a rare and aggressive skin tumor that typically occurs on sun-exposed areas in immune-compromised patients [3]. Oncogenesis and optimal treatment still also remain major issues for experts in this field. This condition is three to five times more aggressive than melanoma. Usually, it is hard to distinguish it clinically, since its color can vary from normal skin color to a red or bluish tone. These carcinomas are rarely tender to the touch and therefore frequently underdiagnosed at first sight, and they often end up being detected at a more advanced stage. The opposite usually occurs with melanoma, since, thanks to its color and dermoscopy, we are sometimes able to detect at a really early stage. The rapid speed at which these tumors grow is the reason why dermatologist is not the first appointment. It is really rare to find this condition at an early stage, as usually by the time it has been discovered it has already spread all over the body.

Merkel cell carcinoma arises from Merkel cells, which are epithelial neuroendocrine cells located in the basal layer of the epidermis where they act as mechanoreceptors, with which they share numerous characteristics (for example, the tumoral mass expresses cytokeratin 20, CD56, and synaptophysin, which are typical of neuroendocrine cells) [1,3]. Melanoma classic signs should not be used for Merkel cell carcinoma, for which instead, the acronym AEIOU (asymptomatic/lack of tenderness, expanding rapidly, immune suppression, older than 5 years old, and ultraviolet-exposed/fair skin) is used [2].

Available evidence has established the role of UV radiation exposure and polyomavirus (PV) infection in the pathogenesis of this neoplasm, with the former being able to explain the coexistence of other sun-related cutaneous tumors with this entity [1,4]. MCC can spread rapidly to local lymph nodes because of its high metastatic rate [3]; however, the spontaneous regression of metastatic tumors has been reported [4]. It follows that total body CT, MRI, or PET-CT are pivotal in performing correct tumoral staging. Nowadays, based on the location of the tumoral mass and local and/or distant involvement, a combination of wide local excision with lymphadenectomy and non-surgical procedures is the currently used therapeutic approach [3]. Radiotherapy may be beneficial, as MCC is radiosensitive, making this a viable option following surgical excision as well as in patients who are not eligible for surgery [3,4,5,6]. Chemotherapy for MCC comprises the association of platinum-based regimens, anthracyclines, and etoposide. Yet, the few reports which address their efficacy have reported low response rates [3]. In our patient chronic sun exposure of immune-senescence have probably played an important role, although we are not able to speculate about any possible correlation with bipolar disorder [7]. According to the rapid and aggressive clinical course of our patient, it is important to underline that the tumor-related mortality rate is still elevated. In the literature, ear involvement is estimated in around 22% of cases [8]; thus, every case should be promptly diagnosed and a combination of different therapeutic strategies applied. 

In conclusion, MCC is a rare but highly aggressive cutaneous neoplasm that must be considered in the presence of a rapidly growing mass. The dermoscopic aspect of our patient was not specific, since there were just polimorphic vessels without milky red areas, although it was useful to suspect the presence of a malignant tumor [9]. Nowadays, if promptly diagnosed, the outlook for this condition is not always hopeless, as there have been major changes in this field. In the past, chemotherapy was the frontline therapy with a temporal benefit, and in the case of recurrence resistance and immunosuppression were major limiting factors. Exciting new class of drugs, including checkpoint inhibitor immunotherapies, are giving promising results. In particular, those obtained with target therapy agents such as Avelumab (anti-PDL1 antibody) [10] may offer a glimmer of hope to these unlucky patients. New trials on combination therapies have to be conducted. However, the scientific understanding of the pathogenesis of MCC is still evolving and further studies to identify new effective therapeutic options are necessary. However, the first step is early diagnosis, which is vital. A dermatological check is therefore the first step for these patients.

## Figures and Tables

**Figure 1 medicina-58-00269-f001:**
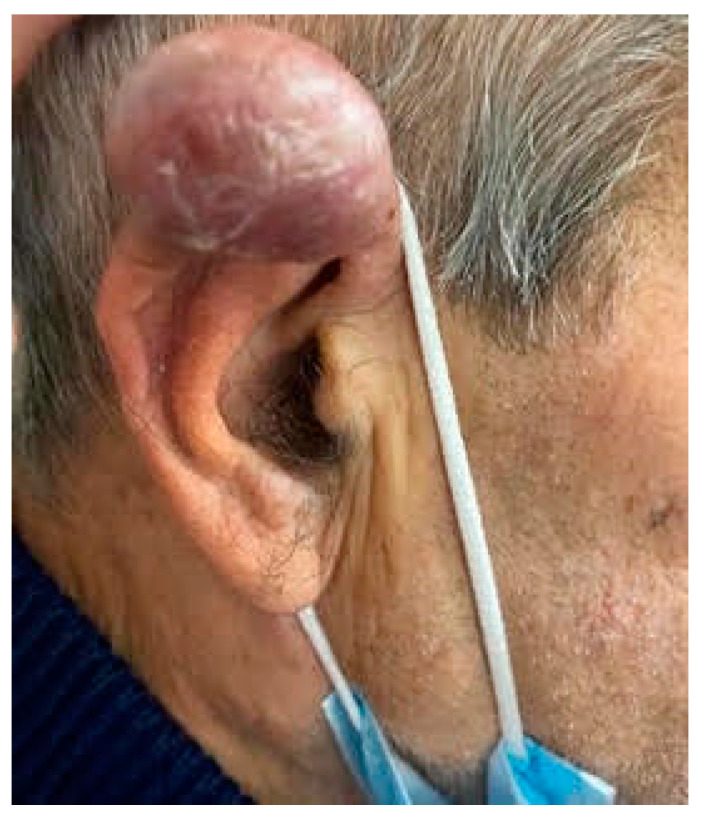
Asymptomatic rapidly growing violaceous nodule on the right helix of the patient.

**Figure 2 medicina-58-00269-f002:**
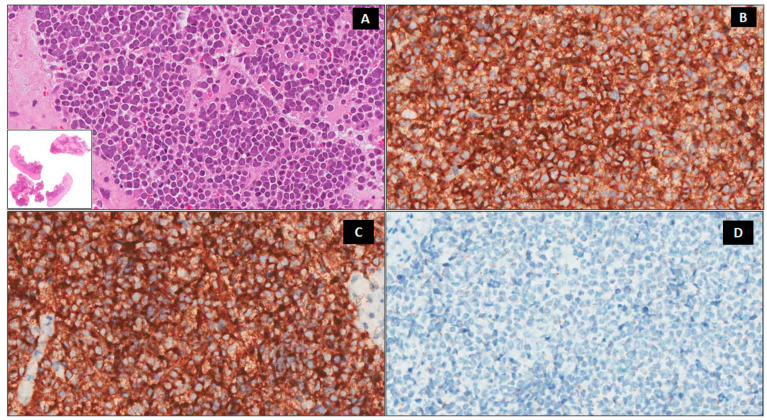
Histologic findings. A predominately dermal blue nodule composed of small–medium-sized round cells with a round nucleus, fine granular chromatin, inconspicuous nucleoli, and scanty cytoplasm (**A**) ×200, Lower insert ×20 hematoxylin-eosin, (H&E). Immunohistochemically, the neoplastic cells are positive for CD56 (**B**) ×200 and synaptophysin (**C**) ×200 and negative for chromogranin A (**D**) ×200.

## Data Availability

Not applicable.

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
