# Peer review of "A Challenging Nodular Lesion of the Ear"

_medicina, 2022, doi:10.3390/medicina58020269_

Round 1
Reviewer 1 Report
This is a very unusual report of Merkel cell carcinoma. The tumor is overall rare and involvement fo the ear is quite exceptional. I think that the paper may be very helpeful for the Clinicians to improve their diagnostic skills.
Only a few suggestions:
please check English language
there are some typing errors in the text; please check and correct
Insert the following artcile in the text and references:
Gioacchini, F. M., Postacchini, V., Simonetti, O., Offidani, A., Magliulo, G., & Re, M. (2013). Merkel cell carcinoma: a systematic review of ENT presentations. European Archives of Oto-Rhino-Laryngology, 270(8), 2191–2199. https://doi.org/10.1007/s00405-012-2283-1Author Response
Thanks for the suggestion we improved english language and typing errors and we inserte the suggested manuscript

Reviewer 2 Report
Thank you to Authors, Editors and Editorial Staff for giving me the opportunity to peer review the current manuscript, which aims to present a case report on Merkel cell carcinoma (MCC). Although MCC is a rare and aggressive tumour representing nowadays a challenge in the dermatologic field, this paper raises some concerns
Nothing new is contributed to scientific knowledge and authors have not generated hypotheses, presented new findings or ideas that could represent an advance for the prevention, diagnosis, prognosis or therapy of this disease. Furthermore, many publications are easily found on this disease. The conduction of the simple following search strategy on MEDLINE/PubMed: ("Carcinoma, Merkel Cell"[mh]) retrieved 2,827 results, being 1,088 of them categorized as case reports according to the pertinent publication type filter.
Finally, the paper should have been much better elaborated, important topics presented in the introduction were not a posteriori discussed and important references are missing.
Author Response
As suggested we tried to improve our manuscript unfortunately the patient died and we were not able to speculate on this case

Reviewer 3 Report
This is a interesting case report regarding the diagnostic challenge of nodular malignant lesions of the skin related to a Merkel cell carcinoma..
I have minor comments to this paper
Line 31. (no clear) delete with all their histopathological variants
Line 31/32 Add Atypical Fibroxanthoma and Kaposi in the differential diagnosis
Case report
Clinical features are very impressive in this case. Dermoscopy has been performed ?
Discussion :
The discussion may be reduced
Line 67. cell carcinoma (CC) or Merkel cell carcinoma MCC ??
Line 82 acronym AEIOU (Asymptomatic/lack of tenderness, Expanding rapidly, Immune suppression, 83 Older than 5p years, and Ultraviolet-exposed/fair skin. A reference is needed.
To give to the reader additional diagnostical key points in the MCC early phase, I suggest to include dermoscopic features of MCC in the discussion. C. Jalilian,1 A.J Clinical and dermoscopic characteristics of Merkel cell carcinoma. British Journal of Dermatology (2013) 169, pp 294–297.
Author Response
Thanks for the possibility to improve our manuscript we followed Your suggestion and added suggested references.
Many thanks in advance
